# Computational Investigation of Conformational Properties of Short Azapeptides: Insights from DFT Study and NBO Analysis

**DOI:** 10.3390/molecules28145454

**Published:** 2023-07-17

**Authors:** Mouna El Khabchi, Mohammed Mcharfi, Mohammed Benzakour, Asmae Fitri, Adil Touimi Benjelloun, Jong-Won Song, Kang-Bong Lee, Ho-Jin Lee

**Affiliations:** 1LIMAS, Department of Chemistry, Faculty of Sciences Dhar El Mahraz, Sidi Mohamed Ben Abdallah University, Fez 30000, Morocco; 2Department of Chemistry Education, Daegu University, Daegudae-ro 201, Gyeongsan-si 38453, Republic of Korea; sjoshua@daegu.ac.kr; 3Climate and Environmental Research Institute, Korea Institute of Science & Technology, Hwarang-ro 14-gil 5, Seoul 02792, Republic of Korea; 4Department of Natural Sciences, Southwest Tennessee Community College, Memphis, TN 38134, USA

**Keywords:** azapeptides, conformational preferences, hydrogen bonds, cis-trans amide bond, β-turn, Asx turn, density functional theory

## Abstract

Azapeptides have gained much attention due to their ability to enhance the stability and bioavailability of peptide drugs. Their structural preferences, essential to understanding their function and potential application in the peptide drug design, remain largely unknown. In this work, we systematically investigated the conformational preferences of three azaamino acid residues in tripeptide models, Ac-azaXaa-Pro-NHMe [Xaa = Asn (**4**), Asp (**5**), Ala (**6**)], using the popular DFT functionals, B3LYP and B3LYP-D3. A solvation model density (SMD) was used to mimic the solvation effect on the conformational behaviors of azapeptides in water. During the calculation, we considered the impact of the amide bond in the azapeptide models on the conformational preferences of models **4**–**6**. We analyzed the effect of the HB between the side-chain main chain and main-chain main-chain on the conformational behaviors of azapeptides **4**–**6**. We found that the predicted lowest energy conformation for the three models differs depending on the calculation methods. In the gas phase, B3LYP functional indicates that the conformers **tttANP-1** and **tttADP-1** of azapeptides **4** and **5** correspond to the type I of β-turn, the lowest energy conformation with all-*trans* amide bonds. Considering the dispersion correction, B3LYP-D3 functional predicts the conformers **tctANP-2** and **tctADP-3** of azapeptide **4** and **5**, which contain the *cis* amide bond preceding the Pro residue, as the lowest energy conformation in the gas phase. The results imply that azaAsx and Pro residues may involve *cis*-*trans* isomerization in the gas phase. In water, the predicted lowest energy conformer of azapeptides **4** and **5** differs from the gas phase results and depends on the calculational method. For azapeptide **6**, regardless of calculation methods and phases, **tttAAP-1** (β-I turn) is predicted as the lowest energy conformer. The results imply that the effect of the side chain that can form HBs on the conformational preferences of azapeptides **4** and **5** may not be negligible. We compared the theoretical results of azaXaa-Pro models with those of Pro-azaXaa models, showing that incorporating azaamino acid residue in peptides at different positions can significantly impact the folding patterns and stability of azapeptides.

## 1. Introduction

Peptides are promising tools with biological and pharmaceutical applications [1,2]. However, their low metabolic stability because of enzymatic degradation, lack of receptor selectivity, and many other drawbacks made it necessary to search for alternatives with the same function as natural peptides but with better properties [3]. These alternatives are called peptidomimetics, in which some structural modifications involve peptides’ backbone or side chains [3,4]. Among peptidomimetics, we focus on azapeptides, a change in the backbone of a peptide by substituting the α-carbon with a nitrogen atom [5,6,7]. This modification confers azapeptides more resistance to physiological degradation and long duration of action [8]. Many studies revealed the importance of incorporating an azaamino acid in the peptide’s sequence to enhance the stability and bioavailability of drug candidates [9,10,11,12,13,14]. In addition, synthetic methods [5,9,11] and biological studies [5,8,9,12,13,14] of azapeptides have been reported. However, the structural behaviors of azaamino acid residue in peptides remain limited.

The conformational preferences of azaamino acid residues have been studied theoretically [15,16,17,18,19,20,21] and experimentally [22,23,24,25,26,27,28,29,30,31]. The results showed that the preferred dihedral angles of the backbone of these azaamino acid residues are in the range of ϕ = ±90° ± 30°, ψ = 0° or ±180° ± 30°, which is appeared to be the polyproline II (β_P_), and β-turn motif (α_R_, α_L_, δ_R_, or δ_L_ conformation) [15,16,17,18,19]. The nomenclature was denoted by Karplus et al. [32]. Our groups and others examined the effect of azaamino acid residue on the structures of peptides [33,34,35,36,37,38]. For example, the azaglycine adopted the polyproline II (β_P_) in azaGly-Pro-Hyp (hydroxyproline) of the collagen [39]; the δ_L_ conformer (ϕ = 90° ± 30°, ψ = 0° ± 30°) in Ac-Phe-azaGly-NH_2_, forming the βII-turn structure; the α_R_ or α_L_ conformer (ϕ = ±90° ± 30°, ψ = 0°) in Ac-Aib-azaGly-NH_2_, adopting βI or βI′-turn [40]. Meanwhile, incorporating azaamino acids destabilized the β sheet secondary structure compared to the parent peptide [41]. The results showed that the conformational preference of azaamino acid residue in peptides should be sequence dependence. However, it is still largely unknown of these questions: why azaamino acid residue adopts a specific conformation in the context dependence, the influence of the side chain of azaamino acid residue on its backbone structure, and the neighboring effect on the conformational behaviors of azapeptides. As a part of the efforts to answer these questions, we previously examined the conformational properties of tripeptides, Ac-Pro-azaXaa-NHMe [Xaa = Asn (**1**), Asp (**2**), Ala (**3**)] using the DFT method, showing that the azaamino acid in this sequence, regardless of the side chain, adopts βII (βII′) turn structure [21]. Additionally, we also suggested the importance of the side-chain and main-chain or main-chain and main-chain HBs found in these model peptides on the stability of the azapeptides. In this work, we extended our ongoing efforts in understanding conformational properties of short azapeptides containing three azaamino acid residues, Ac-azaXaa-Pro-NHMe [Xaa = Asn (**4**), Asp (**5**), and Ala (**6**)] (Figure 1). While previous theoretical studies focused on the dipeptide models [14], we wondered about the position effect of azaamino acid residue in tripeptides. We used the popular DFT functionals, B3LYP and B3LYP-D3 [42], to compare our previous work [21] in the gas phase and water. The solvation effect was calculated with the SMD method [43]. We also investigated the influence of the *cis-trans* isomerization amide bonds on the conformational preferences of model peptides.

Intramolecular HB interactions found in the molecules in the gas phase (Figure 2) were also inspected using Natural Bond Orbital (NBO) theory at the two levels of theory. The strength of the HB in azapeptides models was estimated using the second-order perturbation analysis (E(2) value) [44]. 

## 2. Results and Discussion

The backbone (ϕ and ψ) dihedral angles of the resulting structures for the three azapeptides **4**–**6** are denoted by the nomenclature given by Karplus [32] (Appendix A). Further, the side chain of azaAsn and azaAsp are marked g+(gauche+), g−(gauche−), s+(skew+), and s−(skew−) based on the values of χ_1′_ [45]. Relative energies of the most stable conformations of the three model compounds **4**–**6** calculated at the B3LYP/6-311++G(d, p) and B3LYP-D3/6-311++G(d, p) levels of theory in the gas phase and water are listed in Table 1 and Table 2.

### 2.1. Cis-Trans Isomerization for Minimum Energy Conformations

**Ac-azaAsn-Pro-NHMe (4)**: The conformer **tttANP-1** (α_R_δ_R_(g^+^)), corresponding to the type I of β-turn, and has a *trans-trans* orientation of the amide bond preceding both residues, was found as the lowest energy minimum for azaAsn model in the gas phase calculated with the B3LYP method. However, the same conformation is the second energy minimum at the B3LYP-D3 functional with a relative energy of 0.94 kcal/mol. Interestingly, the lowest energy minimum conformer of azapeptide **4** calculated at the B3LYP functional is found in the X-ray structure containing azaAsn-Pro sequence, showing the presence of a β-Ⅰ turn with all-*trans* amide bonds [46,47]. In inclusion of dispersion correction, B3LYP-D3 functional predicts **tctANP-2** as the lowest energy minimum containing a *cis* amide bond between azaAsn-Pro sequence, which is also predicted to be the most stable conformer in water (Table 2). The second lowest energy minimum in solution calculated with B3LYP functional is **tttANP-12** (δ_L_β_P_(s^+^)) with a relative energy of 0.94 kcal/mol. This relative energy value was also found for the second energy minimum calculated with B3LYP-D3 functional. Another thing to highlight is the *trans* amide bond preceding both Pro and azaAsn found for the five lowest energy minima calculated with the B3LYP functional in the isolated state, except for the second energy minimum in which the amide bond preceding azaAsn is *cis*-oriented (Table 1). However, most of the lowest energy minimum conformers calculated with the B3LYP-D3 functional in the gas phase are *cis*-oriented amide bonds preceding either the two residues or at least one. In the solution state, the five lowest energy minima with B3LYP-D3 functional are all *trans* amide bonds except for the first and the fifth conformers, which have a *cis* amide bond preceding azaAsn residue. The theoretical results suggest that the azaAsn-Pro sequence may also be involved in *cis-trans* isomerization in water.

**Ac-azaAsp-Pro-NHMe (5):** The results of Ac-azaAsp-Pro-NHMe are similar to those of the AzaAsn model. The B3LYP functional predicted **tttADP-1** (α_R_δ_R_(g^+^)) with all-*trans* amide bonds as the lowest energy minimum in the gas phase. The second most stable conformation, **tttADP-2** (α_R_γ’(g^+^)) (ΔE = 1.00 kcal/mol), is also oriented to all-*trans* amide bonds. However, B3LYP-D3 functional predicts the conformer **tctADP-3** as the lowest energy minimum and the conformer **tctADP-4** as the second energy minimum. The results imply that the dispersion interaction may be essential in stabilizing the *cis* amide bond in the azaAsp-Pro sequence. Contrary to the isolated state, both SMD/B3LYP and SMD/B3LYP-D3 functionals predict the conformer **tttADP-9** (δ_L_β_P_(s^+^)) as the lowest energy conformer (Table 2). Similarly, the second most stable conformation in water with the two methods is **tttADP-7**, with all-*trans* amide bonds. However, the third lowest energy minimum **tctADP-3** (βδ_R_(g^+^)) (ΔE = 1.07 kcal/mol) has a *cis* orientation of amide bond preceding Pro residue. The results also imply that the azaAsp-Pro sequence may be involved in *cis-trans* isomerization in water.

**Ac-azaAla-Pro-NHMe (6):** The results of azapeptide **6** significantly differ from those of azapeptides **4** and **5**. Both B3LYP and B3LYP-D3 functionals predict that the conformer **tttAAP-1** (α_R_δ_R_), related to the β-Ⅰ turn structure, is the lowest energy minimum in the gas and water. In addition, the B3LYP function predicts that the four most stable conformers are *trans-trans* amide bonds in the gas phase. However, the third and the fourth energy minimum have *trans-cis* amide bonds with B3LYP-D3 functional. The two most stable conformations calculated by SMD/B3LYP in water have all *trans* amide bonds. 

In contrast, the most stable conformation calculated by SMD/B3LYP-D3 was found to be all *trans* amide bonds followed by two *trans-cis*-*trans* or *cis-trans-trans* conformers (Table 2). The results imply that the side chain of azaAsn and azaAsp may play a vital role in *cis*-*trans* isomer.

### 2.2. Hydrogen Bonds in Ac-azaXaa-Pro-NHMe

Optimized structures of Ac-azaXaa-Pro-NHMe are presented with the denotation of HBs (Appendix A and Appendix A). NBO orbital interactions corresponding to HBs are depicted in the Appendix A, calculated with the B3LYP method. We used the E(2) value to describe the strength of the HB [44].

The most stable conformation in the gas phase at the B3LYP method is **tttANP-1**, which is stabilized with two intramolecular HBs closing C10 and C6 pseudo cycles between carbonyl and N–H groups C^2^=O^1^…H^2^–N^4^ and C^8^=O^4^…H^1^-N^1^, respectively (Figure 3). The C_10_ pseudo cycle is an HB interaction related to the β-Ⅰ turn, which is found in the X-ray structure of Z-azaAsn(Me)-Pro-NHiPr [47]. The HB is made up of three interactions as follows: π_CO_ → σ^*^_NH_ of 1.43 kcal/mol, n_O(σ)_ → σ^*^_NH_ of 1.98 kcal/mol, and n′_O(π)_ → σ^*^_NH_ of 0.81 kcal/mol (Figure 3). The sum ΣE_HB_ of the individual E(2) stabilization energies of the three HB interactions is 4.22 kcal/mol. It is not the strongest interaction, yet **tttANP-1** is the most stable conformation. Moreover, the strongest interaction is that related to the C8 pseudo cycle in **tctANP-7** conformation in which the sum of the individual E(2) stabilization energies, associated with π_CO_, n_O,_ and n′_O_/σ^*^_NH_ overlaps involved in the C.O…HN HB is 9.34 kcal/mol. The same result was found with the B3LYP-D3 method (Appendix A) since the **tctANP-7** conformation has the highest value of the sum of E(2) (9.98 kcal/mol) for the C8 HB pseudocycle. However, the most stable conformation, **tttANP-1**, has a sum value of 5.39 kcal/mol related to the C10 pseudocycle. We notice a slight increase in the values of the sum of the individual E(2) energies calculated with B3LYP-D3 compared to those found with B3LYP. 

The opposite conclusions are drawn in water; the SMD/B3LYP method predicted the **tttANP-12** (δ_L_β_P_) conformer to be the lowest energy minimum in water. This conformer is stabilized with one HB closing a C8 pseudocycle (Appendix A) between the C=O of the main chain and the N-H of the azaAsn side chain. The length of the HB is calculated to be 1.92 Å. Unlike the gas phase, results in water showed that the second most stable conformation is the conformer **tttANP-1**, stabilized with an HB interaction closing a C_10_ pseudocycle, and the HB length is equal to 2.30 Å. We observed the absence of the C_6_ pseudocycle found in the gas phase for the **tttANP-1** conformer. Additionally, the HBs length becomes longer than that in the gas phase.

Unlike the B3LYP functional, the B3LYP-D3 functional predicted the conformer **tctANP-2** to be the lowest energy minimum in gas and solution states. This conformer is stabilized with two intramolecular HBs in the gas phase. One is a backbone-side chain HB closing a C6 pseudocycle, and a backbone HB closing a C5 ring, with the length of HBs estimated to be 2.26 Å for both interactions. The C6 ring was not observed in the solution for this same conformer. However, the length of the C5 HB was slightly larger than that found in the gas phase (2.33 Å). Our result indicates that the side chain of azaAsn in solution may be involved in an intermolecular interaction with water.

The results of compound (**5**) showed few similarities with those of compound (**4**). The most stable conformer in the gas phase with the B3LYP function is **tttADP-1**, with two HBs interactions closing C10 and C6 rings. The C10 HB is related to the βⅠ turn motif for which the sum of the individual E(2) stabilization energies (Appendix A), associated with π_CO_, n_O_, and n′_O_/σ^*^_NH_ overlaps (Figure 4) involved in the CO…HN HB is 3.71 kcal/mol. This turn motif was also found in the X-ray structure of Z-azaAsp(OEt)-Pro-NhiPr [47]. Comparing these results with those observed with B3LYP-D3 functional, we see clearly that the dispersion-corrected B3LYP-D3 functional might not predict the lowest energy minimum; since it is tctADP-3 conformer with one HB closing a C5 pseudocycle (Appendix A) with a sum of E(2) is 1.22 kcal/mol related to the n′_N_ → σ^*^_NH_ NBO overlap of the NH……N backbone HB [48] (Figure 4).

The **tttADP-9** conformer was found as the most stable conformation in water with SMD/B3LYP and SMD/B3LYP-D3 methods. This conformer is stabilized with one HB interaction, a C=O…H–O backbone-side chain HB. We also notice that the length of this type of HB is slightly larger with the dispersion-corrected functional (1.75 Å) than that found with the traditional B3LYP functional (1.64 Å).

The most stable conformation of compound (**3**) in the gas phase with and without dispersion functional is **tttAAP-1**, which is stabilized with an H.B. closing a C10 pseudocycle related to βⅠ turn (Appendix A). The length of such bond is predicted to be 2.16 and 2.08 Å with B3LYP and B3LYP-D_3_ functional, respectively. The HB length has slightly decreased from B3LYP to B3LYP-D3_,_ contrary to azaAsn and azaAsp models. The NBO overlaps of the C10 HB are shown in Figure 5. The sum of E(2) associated with π_CO_, n_O_, and n′_O_/σ^*^_NH_ overlaps calculated with B3LYP functional is 3.62 kcal/mol (Appendix A). While that calculated with B3LYP-D3 functional is 4.15 kcal/mol (Appendix A), which is only made of two π_CO_ and n_O_/σ^*^_NH_ overlaps. 

Results of the azaAla model in the water show significant similarities with those of the gas phase. The **tttAAP-1** conformer is the lowest energy minimum and stabilized with a C_10_ backbone C=O…H–N HB. The length of this HB is found to be 2.32 Å with B3LYP functional and 2.14 Å with B3LYP-D3 functional, which is the same finding as in the gas phase. 

### 2.3. Asx Turn vs. βⅠ-Tun in Ac-azaAsx-Pro-NHMe (x = n or p)

Hydrogen-bonded Asx turns are similar to β turns since both have a C10 pseudocycle. However, in Asx turns, it is the side chain’s carbonyl group of Asn or Asp at the *i +* 1 position, which is hydrogen bonded to NH of the backbone of residue at the *i +* 3 position. Thus, the nomenclature of Asx-turns is the same as β-turns. Four categories were found: type-I, type-II, type-I′, and type-II′ Asx turn (Table 3). This nomenclature is based on the dihedral angles ϕ and ψ of residues *i +* 1 and *i +* 2. Since Asx turns are mimicry of β turns, the dihedral angles ϕ_e_ and ψ_e_ of Asx turn (Figure 6) should resemble those of *i +* 1 in β turns, while ϕ and ψ of residue *i +* 1 of Asx turns should resemble those of residue *i +* 2 in β turns [49,50]. 

Furthermore, the azaAsx-Pro chain contains another C10 pseudocycle similar to an Asx turn, which is related to the HB involving the N^δ^H or O^δ^–H of the side chain of Asn or Asp, respectively, and carbonyl group two residues ahead which will be noted (C10SC). Results of the azaAsn model compound in the gas phase calculated with the B3LYP functional showed that all three conformations adopting the βⅠ-turn motif are more stable than those adopting the Asx turn. However, the most stable Asx turn conformation **tttANP-9** is only 2.89 kcal/mol higher in energy, which means there is a competition between Asx and βⅠ turn secondary structures. The C10 C=O…H–N backbone side chain HB with the sum of E(2) being 5.76 kcal/mol which is associated with n_O_ and n′_O_/σ^*^_NH_ overlaps (Appendix A). On the other hand, the most stable Asx turn conformation with B3LYP-D3 functional is **cctANP-17**, which has a sum of E(2) value of 1.88 kcal/mol related to n_O_ and n′_O_/σ^*^_NH_ overlaps (Appendix A). Moreover, the majority of βI-turn conformations are more stable than the Asx turn, and the **cctANP-16** conformation related to the C10SC interaction is more stable than the Asx turn. The sum of the E(2) value (3.05 kcal/mol) related to the C10 HB of this conformer is higher than that of **cccANP-17**, which means a stronger HB bond interaction. The Asx turn was not reported in the azaAsn-Pro sequence yet. Instead, the N^δ^–H of the side chain of azaAsn is bounded to N^α^ of the same residue [46,47]. Moreover, based on the dihedral angles of a typical Asx turn (Table 3), the most stable Asx turn conformation **tttANP-9** (ΔE = 2.89 kcal/mol for B3LYP; ΔE = 4.52 kcal/mol for B3LYP-D3) is of type-I Asx turn. This Asx turn was also found in Asn-Pro studied with NMR and IR spectroscopies [49]. On the contrary, a study of a database of 500 proteins found that the frequency of occurrence of Asx turn is type-II′, while the most common type is βI turn [50]. These findings were also supported by a recent study of Asx turn in Asn-Ala and Asn-Gly in crystallized protein structures in the PDB [51], revealing that the most common Asx turn in these two sequences are type-II′ and type-II, respectively. For comparison, we calculated the conformational properties of Ac-Asn-Pro-NHMe (**7**, **NP**) and Ac-Asp-Pro-NHMe (**8**, **DP**) at the same levels of theories used here in the gas phase (Appendix A). The most favorable Asn turn of model **7** is **tttNP-02** (ΔE = 1.54 kcal/mol for B3LYP; ΔE = 0.58 kcal/mol for B3LYP-D3), likely type-I Asx-turn. Note that the βI-turn conformer **tttNP-16** (ΔE = 5.60 kcal/mol for B3LYP; ΔE = 3.3 kcal/mol for B3LYP-D3) is predicted to have higher energy than type-I Asx turn conformer **tttNP-02**. We also found that the most favorable Asp turn of model **8** is **tctDP-01** (ΔE = 0.00 for B3LYP and B3LYP-D3). For model **8**, βII′-turn conformer **tttDP-09** (ΔE = 2.51 kcal/mol for B3LYP; ΔE = 4.58 kcal/mol for B3LYP-D3) has higher energy than Asx-turn conformation **tctDP-01** (Appendix A). 

In sum, our results suggest that azaAsn and azaAsp residues in models **4** and **5** could have different folding patterns than natural peptides, and they could also enhance the formation of a type-I Asx turn instead of type-II′ mostly found in peptides. Even though the Asx turn is more stable than the C10SC, which is similar to the Asx turn, most conformers with C10SC are more stable than those adopting the Asx turn, especially with B3LYP-D3 functional. The results are comparable to models Ac-Pro-azaAsn-NHMe (**1**) and Ac-Pro-Asp-NHMe (**2**). The results revealed that the most favorable β-turn structure of azapeptide 1 is the βII(II′) turn, and that of azapeptide **2** is the βII′ turn stabilized by an HB closing a C10 cycle and two other HB cycles, C7 and C5 with B3LYP functional [15]. The results show that depending on the position of Asx residue in the sequence could induce different folding patterns. 

In solution, results showed few similarities regarding the most stable Asx turn conformation with B3LYP functional, which was found to be the conformer **tttANP-9**. However, the **tctANP-14** conformer related to the C10SC turn also has the same relative energy as **tttANP-9** (∆E = 2.95 kcal/mol). The C10SC turn was found to be more stable than Asx turn in solution with the B3LYP-D3 functional since the **tctANP-14** conformer adopting the C10SC turn has a relative energy of 2.07 kcal/mol and the **tttANP-9** conformer related to the Asx turn is 4.20 kcal/mol energy. The result indicates that the side chain of azaAsn in the azaAsn-Pro sequence acts as a proton donor in water.

Similarly, the βI turn structure in the azaAsp-Pro sequence with B3LYP functional is more stable than the Asx turn in the isolated state. This is observed when comparing **tttADP-1** and **tttADP-8** conformers. The latter conformer related to the type-I Asx turn is 3.70 kcal/mol higher in energy than the first one, which adopts the βI turn. It is also noticed that the Asx turn in this sequence is less stable than that found for the azaAsn-Pro sequence. In addition, conformers **cctADP-15** and **tctADP-16** adopt an undefined type of Asx turn stabilized with the C10 cycle. Like the azaAsn model compound, the Asx turn in the azaAsp model compound is more stable than the C10SC motif as we compare the most stable Asx turn conformer **tttADP-8** and the most stable C10SC conformer **cctADP-19** (ΔE = 7.40 Kcal/mol). Additionally, most Asx turn conformations are energetically more favorable than C10SC conformations. Like the B3LYP functional, the B3LYP-D3 functional predicted that the Asx turn is more stable than the C10SC turn in the isolated state. The B3LYP-D3 functionality showed that the **tctADP-16** conformer is the most stable Asx turn with a relative energy of 3.83 kcal/mol, and the sum of E(2) value is equal to 2.47 kcal/mol and associated with n_O_ and n′_O_/σ^*^_NH_ overlaps.

Similar results were found in the solution state since the βI turn is more stable than C10SC and Asx turn conformations. However, the C10SC is more stable than Asx turn with B3LYP functional, conformation **tctADP-22** is the most stable C10SC turn conformation with a relative energy value of 5.15 kcal/mol while conformation **cctADP-15** adopting the Asx turn has a relative energy value of 9.29 kcal/mol. The B3LYP-D3 functional reveals that the C10SC turn is more stable than the Asx turn, with the conformer **tctADP-22** being the most stable C10SC conformer in water with a relative energy of 3.45 kcal/mol.

## 3. Materials and Methods

All calculations were performed using the Gaussian 09/16 programs package. The starting geometries of the three compounds were selected from the previous work [21] and were fully optimized in the gas phase at the B3LYP/6-311++G(d, p) level of theory. These geometries combine all possible intramolecular hydrogen bonds formed in the molecule between acceptor groups C=O or N^α^ and the N–H donor group. The orientation of amide bonds in model azapeptides, which is either *cis* (ω = 0°) or *trans* (ω = ±180°) amide bond, was also assessed. All conformations were found to be true local minima, as was indicated by the absence of imaginary frequencies. These conformations were then reoptimized using the B3LYP-D3 [42] in the gas phase using the same basis set. The solvent effect was examined using the solvation model density (SMD) [43] with and without the dispersion-correction B3LYP functional in water. For comparison, the conformational preferences of peptide models, Ac-Asn-Pro-NHMe (**7, NP**) and Ac-Asp-Pro-NHMe (**8**, **DP**), were investigated using the B3LYP and B3LYP-D3 functionals. The initial structures for peptides **7** and **8** were generated by conformation search implemented in Hyperchem 8.1 [52] with the molecular mechanism. The local minima with ΔE less than 10 kcal/mol were selected and then fully optimized at the B3LYP-D3/3-21G level of theory. The local minima were fully optimized at the B3LYP and B3LYP-D3 functionals with the 6-311++G(d, p) basis set. Intramolecular hydrogen bond interactions found in the molecules in the gas phase were also inspected using natural bond orbital (NBO) theory at the two levels of theory using NBO 3.1 program implemented in Gaussian 16 package [53]. The strength of the hydrogen bond in azapeptides models was estimated using the second-order perturbation analysis (E(2) value) [26]. The NBO overlaps were drawn using the Multiwfn program [54].

## 4. Conclusions

In summary, conformational preferences of three azapeptides models Ac-azaXaa-Pro-NHMe [Xaa = Asn, Asp, Ala] were studied using the B3LYP/6-311++G(d, p) and B3LYP-D3/6-311++G(d, p) levels of theory in isolated and water to examine the effect of side chain’s nature and position effect on the folding patterns of azapeptides. Our result demonstrates that the position of azaamino acid residue largely affects the folding patterns of these compounds. Another finding is the orientation of amide bonds preceding azaamino acid residue and Pro. The most stable conformations of all three compounds in gas and solution states calculated with B3LYP functional are all *trans* amide bonds. Moreover, the stability order of the three secondary structures stabilized with a 10-membered HB in azaAsn and azaAsp models is βI ˃ C10S ˃ Asx for azaAsn-Pro and azaAsp-Pro sequences in water calculated by SMD/B3LYP and SMD/B3LYP-D3, and for the zaAsp-Pro model in the gas phase also calculated by two methods. In contrast, the stability order for the azaAsn-Pro in the gas phase calculated with the two functionals is βI ˃ Asx ˃ C10SC. Note the hat azaXaa-Pro sequence prefers the βI-turn, whereas the Pro-azaXaa sequence prefers the βII-turn structure [15]. Our computational investigation of the conformational properties of short azapeptides demonstrates that the nature of the side chain of azapeptides could largely influence the folding patterns of these components.

## Figures and Tables

**Figure 1 molecules-28-05454-f001:**
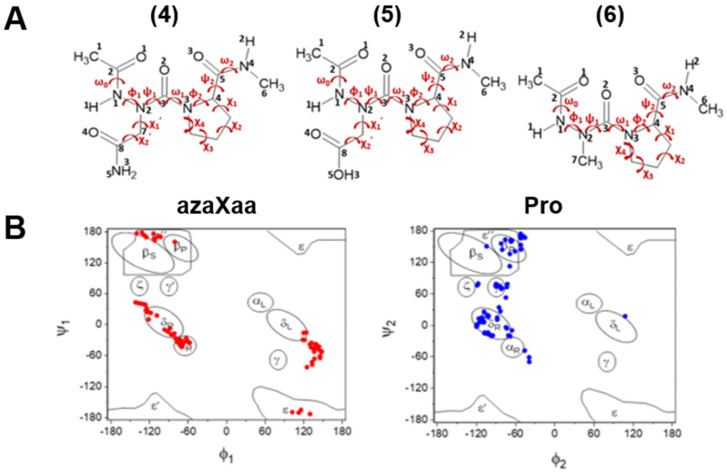
Azapeptide models studied in this work (**A**) Definition of torsional angle parameters of azaamino acid residue-containing tripeptide models, Ac-azaXaa-Pro-NHMe [Xaa = Asn (**4**), Asp (**5**), Ala (**6**)]; (**B**) The backbone (ϕ, ψ) distribution of azaXaa and Pro residues, respectively, calculated at the B3LYP/6-311++G(d, p) level of theory in the gas phase. The nomenclature proposed by Karplus et al. [32] is used: α_R_: right-handed α-helix; α_L_: a mirror image of α-helix; β_S_: region primarily involved in β-sheet formation; β_P_: regions associated with extended polyproline-like helix or β-sheet; γ and γ′: γ and inverse-γ turns; δ_R_: the bridge region; δ_L_: a mirror image of δ_R_ region; ε: the extensive region with ϕ > 0, ψ = ±180°; ε′ and ε′′: mirror images of the two parts of the ε region; ζ: a region associated with residues preceding Pro [19].

**Figure 2 molecules-28-05454-f002:**
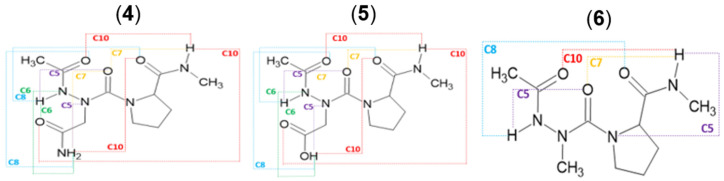
Cyclic motifs formed by hydrogen bonds in Ac-azaXaa-Pro-NHMe [Xaa = Asn (**4**), Asp (**5**), Ala (**6**)].

**Figure 3 molecules-28-05454-f003:**
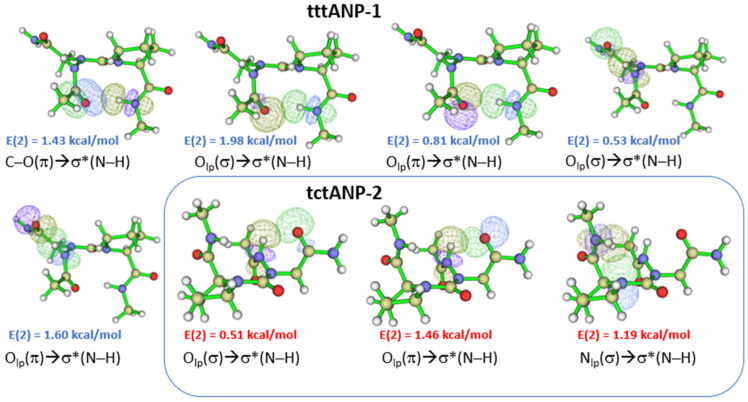
NBO overlaps of **tttANP-1** and **tctANP-2** in the gas phase. ‘t’ and ‘c’ represents ‘*trans*’ and ‘*cis*’ amide bond, respectively. The E(2) value is shown in blue color for B3LYP and red color for B3LYP-D3 functional.

**Figure 4 molecules-28-05454-f004:**
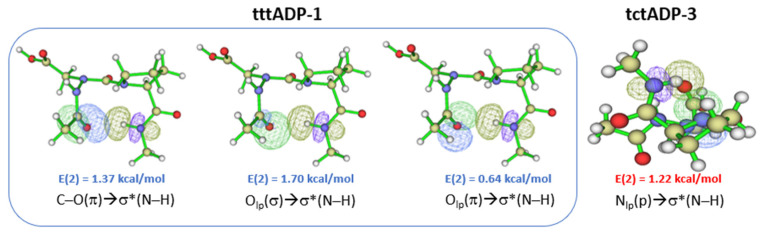
NBO overlaps of the lowest energy conformer **tttADP-1** (@B3LYP) and **tctADP-3** (@B3LYP-D3) in the gas phase. The E(2) value is shown (See Appendix A).

**Figure 5 molecules-28-05454-f005:**
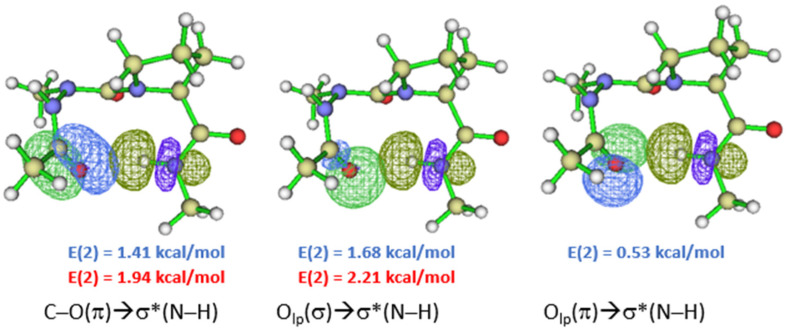
NBO overlaps of **tttAAP-1** in the gas phase. The E(2) value is shown in blue color for B3LYP and red color for B3LYP-D3 functional.

**Figure 6 molecules-28-05454-f006:**
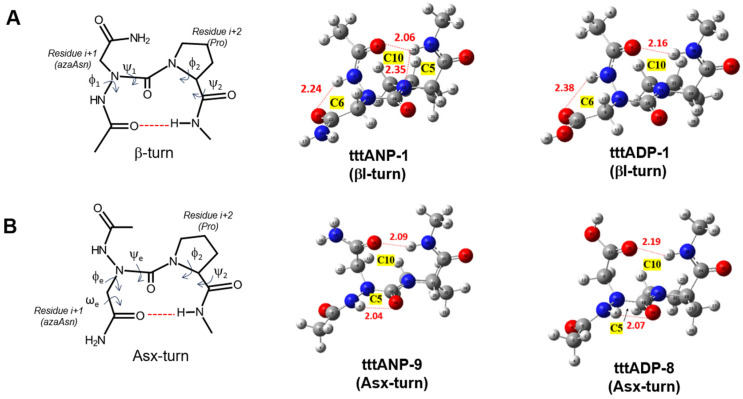
Definition of structure and torsion angles for β-turn (**A**) and Asx-turn (**B**). The representative β-turn and Asx-turn for Ac-azaAsn-Pro-NHMe (**4**, **ANP**) and Ac-azaAsp-Pro-NHMe (**5**, **ADP**) optimized at the B3LYP/6-311++G(d, p) level of theory in the isolated form. HB bond lengths (Å) found in turns are in red color.

**Table 1 molecules-28-05454-t001:** Relative energies in (kcal/mol) of model compounds (**4**), (**5**)**,** and (**6**) at B3LYP (M1) and B3LYP-D3 (M2) functionals with the 6-311++G (d, p) basis set in the isolated form.

Ac-azaAsn-Pro-NHMe			Ac-azaAsp-Pro-NHMe			Ac-azaAla-Pro-NHMe		
	ΔE(M1)	ΔE(M2)		ΔE(M1)	ΔE(M2)		ΔE(M1)	ΔE(M2)
**tttANP-1**	**^a^ 0.00**	0.94	**tttADP-1**	**^c^ 0.00**	0.69	**tttAAP-1**	**^e^ 0.00**	**^f^ 0.00**
**tctANP-2**	0.94	**^b^ 0.00**	**tttADP-2**	1.00	2.70	**tttAAP-2**	0.75	1.13
**tttANP-3**	1.32	2.64	**tctADP-3**	1.07	**^d^ 0.00**	**tttAAP-3**	1.00	2.51
**tttANP-4**	1.38	1.95	**tctADP-4**	1.44	0.38	**tttAAP-4**	1.26	3.70
**tttANP-5**	1.96	2.89	**cttADP-5**	2.07	2.57	**tctAAP-5**	1.44	1.82
**cttANP-6**	2.45	3.07	**tttADP-6**	2.95	3.33	**cttAAP-6**	1.51	2.26
**tctANP-7**	2.64	2.01	**tttADP-7**	3.33	4.39	**cttAAP-7**	1.63	2.20
**tttANP-8**	2.64	3.20	**tttADP-8**	3.70	5.46	**tctAAP-8**	2.26	1.26
**tttANP-9**	2.89	4.52	**tttADP-9**	4.52	5.33	**tttAAP-9**	2.70	4.89
**cttANP-10**	3.07	3.26	**cttADP-10**	4.52	4.58	**tctAAP-10**	2.76	2.20
**cttANP-11**	3.26	2.95	**cttADP-11**	4.58	4.83	**cttAAP-11**	3.51	3.01
**tttANP-12**	4.08	4.89	**cctADP-12**	4.83	3.39	**tctAAP-12**	3.70	5.27
**tctANP-13**	4.27	3.20	**tctADP-13**	5.08	4.89	**cctAAP-13**	3.77	2.01
**tctANP-14**	4.46	3.33	**tttADP-14**	5.65	7.53	**tttAAP-14**	3.89	4.64
**tttANP-15**	4.46	5.65	**cctADP-15**	6.02	4.89	**cctAAP-15**	4.33	2.95
**cctANP-16**	4.52	2.32	**tctADP-16**	6.34	3.83	**cttAAP-16**	4.96	6.46
**cctANP-17**	4.64	3.20	**tctADP-17**	6.78	7.22	**cctAAP-17**	5.52	5.02
**cttANP-18**	4.83	4.96	**cctADP-18**	6.97	4.96	**ttcAAP-18**	6.28	5.58
**tctANP-19**	5.02	3.26	**cctADP-19**	7.40	5.40	**cctAAP-19**	6.34	5.46
**cttANP-20**	5.40	5.71	**tctADP-20**	7.66	6.21	**cccAAP-20**	8.47	6.15
**cctANP-21**	5.46	4.46	**tttADP-21**	7.72	9.04	**tccAAP-21**	10.10	7.66
**cctANP-22**	5.58	3.26	**tctADP-22**	8.03	6.97	**cccAAP-22**	11.92	9.73
**cttANP-23**	5.96	5.77	**cttADP-23**	8.03	7.97			
**tctANP-24**	6.71	7.22	**ttcADP-24**	9.29	8.84			
**tctANP-25**	6.90	5.52	**tccADP-25**	10.10	7.47			
**cttANP-26**	7.59	7.53	**cttADP-26**	10.86	11.11			
**tttANP-27**	7.59	9.66	**cttADP-27**	11.30	10.73			
**ttcANP-28**	7.97	7.59	**cccADP-28**	13.43	9.22			
**tccANP-29**	9.16	6.21	**cctADP-29**	13.43	12.61			
**tttANP-30**	9.79	11.36	**cccADP-30**	16.13	12.74			
**cctANP-31**	9.98	9.48						
**cccANP-32**	10.35	6.34						
**cccANP-33**	14.75	11.80						

^a^ E = −1005.6114 a.u; ^b^ E = −1005.6582 a.u; ^c^ E = −1025.4846 a.u; ^d^ E = −1025.5303 a.u; ^e^ E = −836.8502 a.u; ^f^ E = −836.8898 a.u.

**Table 2 molecules-28-05454-t002:** Relative energies in (kcal/mol) of model compounds **4, 5,** and **6** at the SMD/B3LYP/6-311++G (d, p) (M3) and the SMD/B3LYP-D3/6-311++G(d, p) (M4) levels of theory in water.

Ac-azaAsn-Pro-NHMe			Ac-azaAsp-Pro-NHMe			Ac-azaAla-Pro-NHMe		
	ΔE(M3)	ΔE(M4)		ΔE(M3)	ΔE(M4)		ΔE(M3)	ΔE(M4)
**tttANP-1**	1.13	1.51	**tttADP-1**	3.14	2.70	**tttAAP-1**	**^e^ 0.00**	**^f^ 0.00**
**tctANP-2**	1.38	**^b^ 0.00**	**tttADP-2**	5.21	5.71	**tttAAP-2**	2.01	3.07
**tttANP-3**	3.07	4.52	**tctADP-3**	3.64	1.38	**tttAAP-3**	2.76	4.27
**tttANP-4**	3.01	3.07	**tctADP-4**	5.15	3.58	**tttAAP-4**	2.70	6.46
**tttANP-5**	2.76	3.45	**cttADP-5**	6.84	6.28	**tctAAP-5**	3.26	3.95
**cttANP-6**	4.64	4.96	**tttADP-6**	4.27	3.45	**cttAAP-6**	4.96	5.46
**tctANP-7**	4.27	3.70	**tttADP-7**	2.82	2.45	**cttAAP-7**	5.15	5.71
**tttANP-8**	1.51	1.57	**tttADP-8**	5.40	5.77	**tctAAP-8**	2.01	0.88
**tttANP-9**	2.95	4.20	**tttADP-9**	**^c^ 0.00**	**^d^ 0.00**	**tttAAP-9**	2.64	4.64
**cttANP-10**	6.78	6.59	**cttADP-10**	5.65	5.27	**tctAAP-10**	1.82	2.26
**cttANP-11**	7.15	6.34	**cttADP-11**	7.66	6.90	**cttAAP-11**	2.82	2.32
**tttANP-12**	**^a^ 0.00**	0.82	**cctADP-12**	6.02	3.45	**tctAAP-12**	5.02	6.34
**tctANP-13**	3.33	4.89	**tctADP-13**	5.84	4.33	**cctAAP-13**	4.46	3.07
**tctANP-14**	2.95	2.07	**tttADP-14**	6.09	6.90	**tttAAP-14**	1.69	6.34
**tttANP-15**	4.64	5.84	**cctADP-15**	9.29	7.78	**cctAAP-15**	4.58	3.01
**cctANP-16**	3.64	2.38	**tctADP-16**	6.02	5.08	**cttAAP-16**	3.77	4.96
**cctANP-17**	3.77	2.07	**tctADP-17**	4.71	4.64	**cctAAP-17**	5.52	5.02
**cttANP-18**	3.51	4.02	**cctADP-18**	7.15	4.58	**ttcAAP-18**	4.33	7.22
**tctANP-19**	3.07	2.13	**cctADP-19**	6.02	4.14	**cctAAP-19**	4.46	3.95
**cttANP-20**	4.33	4.58	**tctADP-20**	7.40	5.15	**cccAAP-20**	7.34	6.21
**cctANP-21**	6.78	5.96	**tttADP-21**	5.08	6.84	**tccAAP-21**	7.91	5.58
**cctANP-22**	6.02	4.08	**tctADP-22**	5.15	3.45	**cccAAP-22**	11.04	9.29
**cttANP-23**	5.58	4.02	**cttADP-23**	5.90	4.83			
**tctANP-24**	2.70	3.51	**ttcADP-24**	5.15	10.42			
**tctANP-25**	4.14	4.20	**tccADP-25**	10.17	6.46			
**cttANP-26**	3.89	3.45	**cttADP-26**	8.03	7.53			
**tttANP-27**	4.08	5.84	**cttADP-27**	7.59	6.21			
**ttcANP-28**	10.17	9.79	**cccADP-28**	10.73	6.40			
**tccANP-29**	9.35	6.15	**cctADP-29**	9.91	8.22			
**tttANP-30**	3.07	5.71	**cccADP-30**	12.80	7.72			
**cctANP-31**	6.34	5.52						
**cccANP-32**	8.22	4.64						
**cccANP-33**	12.56	10.10						

^a^ E = −1005.6567 a.u; ^b^ E = −1005.7028 a.u; ^c^ E = −1025.5295 a.u; ^d^ E = −1025.5733 a.u; ^e^ E = −836.8844 a.u; ^f^ E = −836.9238 a.u.

**Table 3 molecules-28-05454-t003:** Backbone dihedral angles of residues at the *i +* 1 and the *i +* 2 positions for the major types of β turns.

Turn	ϕ_i+1_ (°) Asx ϕ_e_	ψ_i+1_ (°) Asx ψ_e_	ϕ_i+2_ (°)	ψ_i+2_ (°)
βI	−60	−30	−90	0
βI′	60	30	90	0
βII	−60	120	80	0
βII′	60	−120	−80	0

## Data Availability

The data supporting reported results can be found from one of the authors (M.E.K).

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
