# Peer review of "Computational Investigation of Conformational Properties of Short Azapeptides: Insights from DFT Study and NBO Analysis"

_molecules, 2023, doi:10.3390/molecules28145454_

Round 1

Reviewer 1 Report

In the manuscript entitled ‘Computational Investigation of Conformational Properties of Short Azapeptides: Insights from DFT Study and NBO Analysis ‘‘computational approaches DFT/ (B3LYP and B3LYP-D3) with the 6-311++G(d,p) basis set was used by the authors to perform all the calculations. The aim was to investigate the conformational preferences of three azaamino acid residues in tripeptide models namely Ac-azaXaa-Pro-NHMe with Xaa= Asn compound (4), Xaa=Asp compound (5) and Xaa= Ala compounds (6). This was because the class of the mentioned compounds (Azapeptides) has gained much attention due to their ability to enhance the stability and bioavailability of peptide drugs. Let me mention that azapeptides are known as inhibitors of cysteine proteases for example, a class of enzymes validated as potential drug targets in the treatment of many diseases. So, investigation of the conformation properties of inhibitors of this class of enzymes is very important. This can help to understand and predict their binding mode with their targets. The paper has been written well with clear conclusions. Beyond the potentiality of the manuscript, I have a few minor concerns that the authors may address.

Comments:

(1)   Why the authors have limited their study to only three compounds?

(2)   No reason why these three compounds have been selected for their study compared to other azapeptides.

(3)   The manner to list reference in the text are different (check for example reference 20 on page 2/13 line 64) check this carefully in the entire text.

Reviewer 2 Report

The manuscript describes a thorough QM study of selected azapeptides. The calculated energies are presented in detail. I have a few suggestions to improve the quality of presentation and analysis.

- It would be of specific interest to compare the structural properties of the azapeptides to those of the corresponding natural peptides even if similar calculations can not be easily found in the literature or performed during a revision. Still, the detailed discussion of the specific steric properties and the underlying differences in the electronic structure between aza- and natural peptides could greatly improve the impact of the paper.   

- On a similar note, the authors use compound numbering 4-6 to keep continuity with their previous work. however, a detailed comparison with compounds 1-3, which could justify this, seems missing.

- I suggest the improvement of the presentation by using more informative names of the conformers (maybe with an explanatory figure on the relationship of the nomenclature and the Ramachandran regions) as well as more informative schemes - for Figure 6 on Asx turns, a scheme on which the Asx turn could be directly compared to a beta-turn, including the H-bonds, could be more informative.

- Minor issues: please use HB or "H.B." consistently. Also check the sentence in lines 289-291 as it contains a text editing error.
